# Insights into the Fe^3+^ Doping Effects on the Structure and Electron Distribution of Cr_2_O_3_ Nanoparticles

**DOI:** 10.3390/nano13060980

**Published:** 2023-03-08

**Authors:** Cledson Santos, John M. Attah-Baah, Romualdo S. Silva Junior, Marcelo A. Mâcedo, Marcos V. S. Rezende, Robert S. Matos, Ştefan Ţălu, Dung Nguyen Trong, Simone P. A. da Paz, Rômulo S. Angélica, Nilson S. Ferreira

**Affiliations:** 1Department of Physics, Federal University of Sergipe, São Cristóvão 49100-000, SE, Brazil; 2Laboratory of Corrosion and Nanotechnology (LCNT), Federal University of Sergipe, São Cristóvão 49100-000, SE, Brazil; 3Amazonian Materials Group, Federal University of Amapá, Macapá 68902-280, AP, Brazil; 4The Directorate of Research, Development and Innovation Management (DMCDI), Technical University of Cluj-Napoca, 15 Constantin Daicoviciu St., 400020 Cluj-Napoca, Romania; 5Faculty of Physics, Hanoi National University of Education, 136 Xuan Thuy, Cau Giay, Hanoi 100000, Vietnam; 6Institute of Geosciences, Federal University of Pará, Belém 66075-110, PA, Brazil; 7PPGCA, Universidade Federal do Amapá, Macapá 68902-280, AP, Brazil

**Keywords:** chromium oxide, Cr_2_O_3_, tapioca, nanoparticles, electron distribution

## Abstract

Herein, we carefully investigated the Fe^3+^ doping effects on the structure and electron distribution of Cr_2_O_3_ nanoparticles using X-ray diffraction analysis (XRD), maximum entropy method (MEM), and density functional theory (DFT) calculations. We showed that increasing the Fe doping induces an enlargement in the axial ratio of *c*/*a*, which is associated with an anisotropic expansion of the unit cell. We found that as Fe^3+^ replaces Cr in the Cr_2_O_3_ lattice, it caused a higher interaction between the metal 3*d* states and the oxygen 2*p* states, which led to a slight increase in the Cr/Fe–O1 bond length followed by an opposite effect for the Cr/Fe–O2 bonds. Our results also suggest that the excitations characterize a well-localized bandgap region from occupied Cr *d* to unoccupied Fe *d* states. The Cr_2_O_3_ and Fe-doped Cr_2_O_3_ nanoparticles behave as Mott–Hubbard insulators due to their band gap being in the *d*−*d* gap, and Cr 3*d* orbitals dominate the conduction band. These findings suggest that the magnitude and the character of the electronic density near the O atom bonds in Cr_2_O_3_ nanoparticles are modulated by the Cr–Cr distances until its stabilization at the induced quasi-equilibrium of the Cr_2_O_3_ lattice when the Fe^3+^ doping values reaches the saturation level range.

## 1. Introduction

Fascinating characteristics presented as physical and chemical properties that are mostly dependent on the size and shape oxide nanoparticles have motivated several researchers to focus their research in this direction over the past three decades. Recently, several studies have pointed out that a wide assortment of technological applications require transition metal oxide nanoparticles with a controllable mean size, narrow distribution, and specific morphology [1]. For instance, oxide nanoparticles whose properties depend closely on their size, morphology, and surface area activity have been widely reported. Some of these include oxides classified in the corundum-type structure (i.e., R_2_O_3_: R = Fe, Ti, V, Cr, Al), which show a variety of interesting magnetic and electronic phenomena [2,3,4], e.g., weak ferromagnetism in α-Fe_2_O_3_ [5,6,7], magnetoelectric coupling in Cr_2_O_3_ [8,9], and the metal–insulator transition in V_2_O_3_ and Ti_2_O_3_ [10,11,12,13]. Among these corundum-type structure oxides, Cr_2_O_3_ is one of the most important *p*-type semiconductor transition metal oxides to have both Mott–Hubbard insulator results and charge transfer semiconducting properties, with a band gap > 3 eV and high intrinsic resistivity of ρ∼10^12^ Ω.cm at 300 K [14], as the valence band maximum and conduction band minimum are a mixture between Cr 3*d* states and O 2*p* states [15,16]. Furthermore, Cr_2_O_3_ is an antiferromagnetic ordering material [17] that has historically been sported as one of the first oxides to present a linear magnetoelectric effect, giving rise to ferroelectric polarization induced by an external magnetic field [18,19,20]. Despite the active research in the field of magnetism, Cr_2_O_3_ is also useful for a wide variety of applications, including methanol synthesis [21], oxygenation catalyst [22], hydrogen storage [23], gas sensor [24], adhesion promotor, protective coating, and solid oxide fuel cell anode material applications [25]. However, most of these highlighted applications require non-trivial physical and chemical properties that are unexpected in the Cr_2_O_3_ bulk, making the synthesis of Cr_2_O_3_ nanostructures an important challenge. More recently, Cr_2_O_3_ has received considerable attention as a *p*-type transparent conductor because it is less sensitive to interference and it easier to control the grain size during preparation than with other traditional oxides, e.g., SnO_2_ [26], CuAlO_2_ [27], and CuO_2_ [28]. However, substitutional doping by replacing Cr on the Cr_2_O_3_ lattice with different metals is crucial to improve its electrical conductivity and retain its transparency, allowing the material to efficiently act as a transparent conducting component [29,30,31,32,33].

Much effort has been recently made to find out which dopant cation and valence states are most efficient in elevating the conductivity of Cr_2_O_3_ by doping the Cr^3+^ sites. Some of the most popular metal dopants that have been considered in the literature are Ni^2+^ [34], Ti^4+^ [35,36], and Mg^2+^ [37,38,39]. Nonetheless, experiments and calculations have shown that lower valence doping of 2+ cations in Cr_2_O_3_ may induce the formation of charge-compensating vacancy, resulting in electrons being released into the Cr_2_O_3_ lattice and filling the oxygen holes [31,32,40,41,42]. Nevertheless, isovalent doping has positive effects on the electronic properties of other corundum-type structures, such as Fe_2_O_3_ [43], V_2_O_3_ [44], and Al_2_O_3_ [45]. For instance, the substitutional doping of the isovalent Al^3+^ cation at the Fe_2_O_3_ sites removes a second lattice electron, enhancing the formation of oxygen vacancies and the migration of oxygen throughout the host material, resulting in local geometry changes as well as a change in the electronic properties [46].

Notwithstanding, the changes in the electronic structure of Cr_2_O_3_ via aliovalent doping at Cr^3+^ sites have been a matter of recent debate, both experimentally and theoretically; a comprehensive understanding of the mechanism behind the cation doping distribution, the formation of oxygen vacancies, and their interactions to improve the key properties of this oxide is needed. Some authors [47,48] have reported that the replacement of Cr^3+^ by aliovalent cations in the Cr_2_O_3_ lattices results in an increasing concentration of oxygen vacancies due to changes in the electronic structure and lattice distortion. However, most of the scientific interest so far has been directed at the electronic structure change-assisted formation of oxygen vacancies. On the other hand, few previous experimental investigations have reported on metal–metal distance changes induced by lattice distortion in corundum-type structures [2,49]. This makes us question whether a physical factor is favorable for the O vacancy formation other than the anisotropic lattice strain induced by trivalent ion doping. Therefore, a comprehensive understanding of the influence of changing Cr–Cr distances in the lattice distortion character of Cr_2_O_3_ is also of great importance to clarify the effects of dopants on the electronic modification of this material. To this end, we synthesized Fe-doped Cr_2_O_3_ nanoparticles using a cost-effective and environmentally friendly cassava-starch-assisted sol–gel approach [50].

Even though Fe^3+^ is an isovalent ion that may disturb the conduction band by creating a localized positive charge or oxygen vacancies around Cr, many investigations have not focused on the Fe^3+^ doping in the Cr_2_O_3_ lattice. Theoretically, Fe^3+^ doping in the Cr_2_O_3_ lattice has been addressed so far through a few studies of density functional theory (DFT) that have reported on the structure, relaxation, and electronic properties of Fe-doped Cr_2_O_3_ [51,52]. Nonetheless, experimental studies reporting on the structural and electrical properties of Fe-doped Cr_2_O_3_ are scant. Recently, Goel et al. [53] synthesized Fe-doped Cr_2_O_3_ nanoparticles and reported that the incorporation of Fe^3+^ in the Cr_2_O_3_ lattice resulted in unit cell volume expansion and tensile stress on the surrounding crystal lattice. Their results also showed that the optical band gap values were around 3.44–3.09 eV for Fe contents ranging from 0 to 4 wt%, while the resistivity value was decreased due to the carrier concentration increases caused by Fe-doping-induced defects. Despite these exciting results, no one, to the best of our knowledge, has hitherto reported experimental or both theoretical and experimental studies focused on the crystal structure and conductivity performance of Fe^3+^-doped Cr_2_O_3_ nanoparticles. Actually, when Fe is substitute for Cr, the Fe site is expected to be different from the Cr site in Cr_2_O_3_ due to the slightly smaller ionic radius of chromium (*r*^3+^_Cr_ = 0.605 Å) [54] compared to that of Fe (*r*^3+^_Fe_ = 0.645 Å) [55] in octahedral sites. Consequently, an understanding the local environment around the chromium atom is needed to explain the evolution of the physical properties concerning the chromium content *x* in Cr_2−*x*_Fe*_x_*O_3_. For instance, understanding how lattice distortions induced by Fe^3+^ doping in Cr_2_O_3_ influence the character of the Cr–Cr distance, and most importantly whether the electron density can stabilize at the critical Cr–Cr separation point, remains an open issue. Thus, investigations of Cr–Cr repulsion across the shared face of the distorted trigonal antiprisms in the 3-fold axis of the Cr_2−*x*_Fe*_x_*O_3_ crystal lattice are fundamental to clarify the Fe^3+^ doping content’s effect on the short and long Cr/Fe–O covalent bonds, and how it affects the degree of a crossover between antiferromagnetic states.

To date, we have used Rietveld refinements and differences in Fourier synthesis from X-ray data to characterize the crystal structure and electron density distributions in Cr_2_O_3_ crystals, respectively. Furthermore, it is worth mentioning that possible errors from spurious spikes of positive electronic density values that do not correspond to atoms in the structure or due to the presence of “non-physical” local minima with negative density values are recognizable factors of electron density maps. This information loss (experimental noise) issue has been overcome by applying extinction corrections based on the maximum entropy method (MEM) for electron density values [56]. The MEM is a well-established high-resolution tool based on the probability statistical approach theory, which has been used for the precise reconstruction of the electron density maps from X-ray diffraction data [56,57]. The MEM requires a minimum of information, such as lattice parameters and structure factors retrieved from the Rietveld refinements, to construct the electron density distribution in the unit cell. Therefore, it is a suitable technique for analyzing the nature of the bonding behavior and the distribution of electrons in the bonding region of interest. Furthermore, more detailed insights into this fundamental issue can be achieved using density functional theory (DFT) calculations, which are also suitable for providing accurate results for the electron density maps, density of states, band structure, electrical polarization, and atomic structure.

Herein, we discuss the effect of the iron content on the spatial distribution of electronic charges and its consequences for the structural properties of greenly synthesized Fe-doped Cr_2_O_3_ nanoparticles through an experimental study using an XRD analysis and the MEM. The main goal of this study was to investigate how Cr_2_O_3_ lattice distortions induced by Fe^3+^ doping influences the character of the Cr–Cr distance, and most importantly whether the electron density can be stabilized at the critical Cr–Cr distance point. To shed more light on the Fe^3+^ doping effects on the charge distribution and covalent nature of Cr–O bonds, the chemical bonds and electronic structure of the Cr_2_O_3_ lattice were investigated via the electron density, density of states, and band structure based on DFT total energy values and compared to those obtained from the experimental analysis.

## 2. Materials and Methods

### 2.1. Synthesis

The Cr_2−*x*_Fe*_x_*O_3_ (0.0 ≤ *x* ≤ 0.18) nanoparticles were synthesized using a cassava-starch-assisted sol–gel method [50]. In the typical synthesis process, 5.00 g of cassava starch (tapioca) was first mixed in 150 mL of distilled water under constant stirring. Then, appropriate proportions of analytical-grade Cr(NO_3_)_3∙_9H_2_O (Sigma-Aldrich, >99.9%) and Fe(NO_3_)_3_·9H_2_O (Sigma-Aldrich, >99.9%) were thoroughly mixed in the tapioca solution while stirring it to gain a homogeneous precursor solution. After 1 h of stirring, the solutions were kept at 70 °C for 1 h, and then the obtained gel was dried at 100 °C overnight. The swelled xerogels were ground and calcined at 500 °C for 1 h in a furnace to obtain the pure and doped Cr_2_O_3_ nanoparticles.

### 2.2. Characterization

The X-ray diffraction (XRD) patterns were collected with an X’Pert Pro 3 MPD (PW 3040/60) PANalytical X-ray diffractometer, with a PW3050/60 (θ-θ) goniometer and model PW3373/00 Cu anode ceramic X-ray tube (*K_α1_* = 1.540598 Å) with a long thin focus, Ni-*K_β_* filter, and PIXCEL1D detector (for real-time multiple scanning) in scanning mode. The following instrumental conditions were used: scan 5° at 100° 2θ, 40 kV, 40 mA, step of 0.01° in 2θ and time/step of 20 s, fixed slit of 1/4° and 1/2° anti-scattering, mask of 10 mm.

The powders for the crystalline phase were identified by comparing the experimental XRD patterns to standards from the International Center for Diffraction Data (ICDD) database using X’Pert Highscore software (PANalytical). The instrumental resolution function for the broadening effects was obtained using a silicon standard (Si SMR640d NIST) by fitting the peak profile with WinPlotr software [58]. The angular dependence of the peak full width at half maximum (FWHM) was described by Cagliotti’s formula [59].

The Rietveld refinement [60] of the whole XRD patterns was carried out using the FullProf program [61], with the space group and unit cell parameters found in the indexing process. The pseudo-Voigt function modified by the Thompson–Cox–Hastings [59] function was used to fit the various parameters to the data points, such as a single scale factor, a single zero shift, the fourth-order polynomial background, three cell parameters, five shapes and widths of the peaks, a single overall thermal factor and two asymmetric factors, the occupancies of individual atomic sites, and the pseudo-Voigt profile function. All parameters were refined using an interactive least-squares method [60] to minimize the difference between the experimental and calculated profiles. The electron density distributions were calculated using the MEM using the structure factors obtained from the Rietveld analysis. These calculations were performed using Dysnomia software [62] using a high-resolution grid (by partitioning the unit cell into 128 × 128 × 128 pixels) and using the limited-memory Broyden–Fletcher–Goldfarb–Shanno (L-BFGS) optimization algorithm [63]. The uniform prior density was used in all cases by dividing the total number of electrons by the volume of the unit cell. The two-dimensional images of the electron density distributions were drawn using the VESTA software [64].

### 2.3. First-Principles Calculations

To better understand the evolution of the electronic structure of Cr_2_O_3_ with different Fe doping contents, we performed spin-polarized DFT calculations with a full-potential linearized augmented plane wave (FP-LAPW) method [65] implemented in the WIEN2K computational code [66]. This method partitions the unit cells into non-overlapping spheres positioned at each atomic site (with *R_MT_* radii), where the electron wave functions, charge density, and crystalline potential are expanded as a combination of atomic-like orbitals, and into the interstitial region, for which the expansion takes place through plane waves. The radius of the atomic spheres chosen for both Cr and Fe was 1.9, while for O, we used a radius of 1.64 (in a.u.). The partial waves inside the atomic spheres were developed up to *l_max_* = 10, and a cut-off limited the number of plane waves in the interstitial region at *k_max_* = 7.0/R_MT_(O). The charge density was Fourier-expanded with *G_max_* = 14 Ry, and the cut-off energy for separating the core and valence electronic states was −6.0 Ryd. Electronic calculations were performed with a mash of 63 *k*-points in the *k*-space integration of the irreducible part of the Brillouin zone. Exchange–correlation (XC) effects were taken into account by applying PBEsol-generalized gradient approximation [67] to optimize the lattice parameters and relax all atomic positions, using the Becke–Johnson potential modified by Tran and Blaha (TB–mBJ) [68] for the investigation of the electronic structure associated with the density of states, electron density map, and Bader’s topological analysis of the systems. Finally, the self-consistent calculations were successfully converged as energy values with the convergence criterion of 10^−5^ Ryd. For the simulation, we used the experimental lattice parameters *a* = *b* = 4.956 Å and *c* = 13.593 Å for Cr_2_O_3_ as determined in this work. The Fe-containing system was prepared upon the fully relaxed Cr_2_O_3_ structure, in which two Fe ions replaced two Cr ions. Because the simulated Cr_2_O_3_ is characterized by a rhombohedral structure (space group number 167) containing 30 atoms per unit cell, the substitution Fe → Cr represents a defect concentration of 2/30, which is large when compared to the real systems but reasonable for computational treatments. Regarding the magnetic properties, the most stable magnetic order of the Cr_2_O_3_ compound is G-type antiferromagnetic, where each magnetic spin of the Cr ions alternates in an up and down projection along with the *c*-axis [15,69]. The magnetic cell containing Fe as a dopant was built using the same design. In other words, we inserted the Fe dopant into the Cr_2_O_3_ lattice so that the replacement maintained the same magnetic order as the initial structure. This was supported by recent theoretical studies [51,70] that showed that the magnetic order of the Fe-doped system depends both on the Fe content in the host matrix and on the position where the Fe enters. These studies also concluded that the system maintains the G-type antiferromagnetic order for low Fe ion concentrations. The electronic bulk properties of pure and Fe-doped Cr_2_O_3_ materials were simulated based on carefully determined crystalline structures. We started the calculations by relaxing the lattice parameters and all atomic positions in view to mitigate the stresses in the unit cells and reach values that corresponded to the minimum energy level. The changes in the character of the chemical bonds and the local structures promoted by the presence of Fe defects were investigated in terms of the atoms in the molecules used in Bader’s quantum topological analysis (QTAIM) of the electronic density $\varrho \left(\vec{r}\right)$ [71]. This analysis approach divides space into uniquely defined regions of volumes Ω (called atomic basins) that contain exactly one nucleus by applying the zero-flux condition $\vec{\nabla }\cdot \varrho \left({\vec{r}}_{c}\right)=0$. In this context, the values of $\varrho \left(\vec{r}\right)$ calculated at the bond’s critical point (bcp) ρ_b_ (a type of special point ${\vec{r}}_{c}$ to which their occurrence between two atoms indicates that they are chemically bonded) provides information about the degree of ionicity or the covalent nature of the bonds. It is also possible to draw inferences about the oxidation state of the atoms by calculating the electronic charge contained in each atomic basin (referenced as Bader’s charges *q*(Ω) in this work). Furthermore, the bond valence sum was computationally evaluated.

## 3. Results and Discussion

Figure 1a–d shows the observed, calculated, and differential patterns of XRD Rietveld refinements for the Cr_2−*x*_Fe*_x_*O_3_ (0.0 ≤ *x* ≤ 0.18) nanoparticles. All samples present only diffraction peaks corresponding to the crystallographic planes (012), (104), (110), (006), (113), (202), (024), (116), (122), (214), and (300) allowed for Cr_2_O_3_, with a rhombohedral structure and the space group of R-3c (ICSD#202619). No additional diffraction peaks from any impurities or a second phase were observed, which indicates an important contribution because the tapioca-assisted sol–gel method is simpler, cost-effective, and has a lower environmental impact compared with the hydrothermal [72], chemical precipitation [73], and combustion synthesis [74] methods. The Rietveld refinement analysis was performed by assuming the R-3c (167) space group for a rhombohedral-like structure. The refined parameters, such as the occupancy, atomic functional positions, lattice parameters, unit cell volume, selected interatomic distances, atomic coordinates, occupancy, and thermal parameters of Cr_2−*x*_Fe*_x_*O_3_ (0.0 ≤ *x* ≤ 0.18) nanoparticles are summarized in Appendix A. The fitting parameters *R_p_* = 5.40–6.20%, *R_wp_* = 6.82–7.90%, *R_exp_* = 6.19–6.98%, and *χ*^2^ = 1.16–1.34 indicate a good agreement between the refined and observed XRD patterns for the rhombohedral Cr_2_O_3_ phase.

The next step was to understand the effect of substituting Fe^3+^ for Cr^3+^ in the Cr_2_O_3_ lattice from the Rietveld refinement analysis. The calculated axial ratio 2.74105(5) ≤ *c*/*a* ≤ 2.74240(7) of the Fe-doped Cr_2_O_3_ nanoparticles was somewhat higher than the *c*/*a* = 2.7333 ratio of an ideal hexagonal close-packed bulk Cr_2_O_3_ [75]. The refinement was reached after the Fe was initially placed on a general xyz site, at x = 0, y = 0, and z = 0.34702–0.34783, near the Cr site (12c) positions. Furthermore, the O at the 18e site showed a significant decrease in the x-position from 0.31465 to 0.30669 with an overall variation of ∆x/x = 0.6% as the Fe doping increased from 0 to 0.18 (see Appendix A). We can also see that the unit cell volume increased continuously from *V* = 289.32(2) Å^3^ to 289.82(5) Å^3^, with an overall variation of ∆*V*/*V* = +0.2% as the Fe doping increased from *x* = 0 to *x* = 0.18. Furthermore, the refinement of the structure revealed the XRD broadening lines of the Cr_2−*x*_Fe*_x_*O_3_ (0.0 ≤ *x* ≤ 0.18) nanoparticles, showing anisotropic line broadening with a clear tendency for the (0*k*0) reflections being less broadened than the others, indicating that diffracting domains are substantially anisotropic and with a small microstrain contribution. The Gaussian size and Lorentzian strain broadenings were found to be negligible. Therefore, the microstructural parameters were evaluated, describing the broadening of the Lorentzian part of the diffraction profile due to the size effect and that of the Gaussian part due to the strain effect. Thus, the anisotropy of the size broadening profile was described by a spherical harmonics model according to the following formula [76]:(1)$$\beta_{hkl}=\frac{\lambda }{D_{hkl}\cos\theta }=\frac{\lambda }{cos\;\theta }\underset{lmp}{\sum }a_{lmp}y_{lmp}\left(\Theta_{hkl}\Phi_{hkl}\right)$$

In this model, $\beta_{hkl}$ is the integral breadth of the reflection (*hkl*). Furthermore, the $y_{lmp}\left(\Theta_{hkl}\Phi_{hkl}\right)$ term represents the real spherical harmonics (where $\Theta_{hkl}$ and $\Phi_{hkl}$ are the polar and azimuthal angles of the vector [*hkl*], respectively, concerning a Cartesian crystallographic frame) and $a_{lmp}$ represents the refinable coefficients, depending on the Laue class (in our case, the obtained structure belongs to the Laue group R-3c) [77]. The average crystallite size (*D*) along each reciprocal lattice vector and the microstrain are calculated after refinement of the $a_{lmp}$ coefficients, when the parameters (U,V,W)*_instr._* are fetched by the program from an external instrumental resolution function file. The anisotropy of the strain broadening was modeled from the variance of a quartic form *M_hkl_* in the reciprocal space according to the following equation [78]:(2)$$\sigma^{2}\left(M_{hkl}\right)=\underset{HKL\left\{H+K+L\right\}}{\sum }S_{HKL}h^{H}k^{K}l^{L}$$
where the number of refined *S_HKL_* coefficients depends on the crystalline symmetry. Herein, it is worth mentioning that the anisotropy is the deviation from the average distribution of the grain size along with different directions of the reciprocal lattice. Thus, the standard deviation for the global average apparent size is a measure of the degree of anisotropy, not an estimated error.

The average crystallite size determined using the spherical harmonics method and the microstrain given from Equation (1) for the Cr_2−*x*_Fe*_x_*O_3_ (0.0 ≤ *x* ≤ 0.18) nanoparticles are summarized in Appendix A. The average crystallite size increases from 22.9 ± 2.8 nm (*x* = 0.0) to 75.5 ± 7.7 nm (*x* = 0.18) as the Fe doping content increases (Figure 2b). Thus, the anisotropy rapidly increased from 2.8 nm at *x* = 0.0 to 7.7 nm at *x* = 0.18, corresponding to approximately 36% of the average size. Herein, it is worth highlighting that the axial ratio of *c*/*a* became larger with a crystallite size reduction, which was associated with an anisotropic expansion of the unit cell, followed by a significant elongation of the *c*-axis length (∆*c*/*c* = +0.03%) relative to the *a*-axis (∆*a*/*a* = +0.07%). The symmetry of the structural units with decreasing sizes was opposite to what was suggested by Ayyub et al. [79]. However, it interestingly corroborated the inversely proportional trend of the apparent crystallite size and microstrain values. Factually, the maximum strain was lower for the Cr_1.82_Fe_0.18_O_3_ nanoparticle of <ε> = 1.4 × 10^−4^ (%) and higher for the Cr_2_O_3_ nanoparticle of <ε> = 11.2 × 10^−4^ (%). This higher value for the anisotropy corresponds to less spherical grains or to a short distribution of crystallite sizes in the sample volume. This suggests that the higher content of Fe^3+^ substituted for Cr^3+^ in the Cr_2_O_3_ lattice may change the crystallization rate of the solid phases. As a result, crystallographic domain aggregation is favored by an effect of surface energy minimization, promoting the occurrence of viscous flow in the early sintering stage. Thus, the Cr_2_O_3_ powder with a higher Fe content is a rather larger crystallite within a dense and compact structure, in which the anisotropy is much less accentuated, resulting in a lower average microstrain.

Hence, we can notice in the first approximation that the octahedral site distortion seems directly linked to the microstrain, which is directly governed by the Fe-doping-induced crystal growth. The higher distortion of the octahedral sites for the Cr_1.88_Fe_0.12_O_3_ nanoparticle is because of a decrease in the microstrain. According to the Cr–Cr distance values reported in Table 1, the corresponding Cr–Cr distance in the (110) plane versus the Fe doping content of the Cr_2−*x*_Fe*_x_*O_3_ (0.0 ≤ *x* ≤ 0.18) nanoparticles is illustrated in Figure 2c. The solid red curve indicates the fit of the data to the theoretical curve for a logistic function, as described in Equation (3):(3)$$d_{\left(Cr-Cr\right)}=d_{1}+\left[(d_{1}-d_{2})/(1+{\left(\frac{x_{Fe}}{x_{mFe}}\right)}^{n}\right]$$
where the constant initial growth$\;d_{1}=4.9573\;Å$ and saturation level $d_{2}=4.9603\;Å$ represent the Cr–Cr distances for *x* = 0 and *x* = 0.18, respectively. The growth rate *n* = 5 and the Fe doping midpoint $x_{mFe}=0.089\left(4\right)$ are fitted parameters. It can be supposed that at the Fe doping midpoint, the higher microstrain value related to the smallest long Cr/Fe–O bond length does not allow a stable configuration to be reached with the higher Madelung energy for the Cr_2_O_3_ lattice. This point will be further considered in the next discussion related to MEM investigations.

The modeled crystallite shapes related to the symmetry properties of the distribution of columns of scattering centers in crystalline domains of the Cr_2−*x*_Fe*_x_*O_3_ (0.0 ≤ *x* ≤ 0.18) nanoparticles are shown in Figure 3. The two-dimensional projections of the average crystallite size are visualized on the directions [010], [001], and [100], corresponding to the XZ, YX, and ZY planes, respectively, using the GFourier Program [80]. We can see that the projected crystals present different growth features on the x and y planes. The maximal apparent crystallite sizes of 229, 310, 435, and 773 Å in the [010] direction is almost equivalent to 251, 435, 474, and 875 Å along [001] and 181, 292, 414, and 682 Å in directions perpendicular to direction [100], respectively. Additionally, different anionotropic growths can be observed for the projected crystals in the x, y, and z directions shown in the three-dimensional images. These crystallites tend to present a pitanga-like shape, which is a common growth feature. Similar behavior has been observed previously for ZnO nanocrystals [81]. However, it is worth mentioning that the relation between the crystallite size and the concentration of the defects was remarkable for the sample with *x* = 0.12, suggesting that the Fe content increasing up to this value induces an increase in the symmetry of the surface atoms or decreases the number of unsaturated surface states of Cr^3+^/Fe^3+^. This also resulted in an anisotropic contraction of the unit cell, followed by a significant decrease in the *c*-axis length relative to the *a*-axis, resulting in increased symmetry of the structural units (see Figure 2a). Based on the above results, we can confirm that the lattice expansion and contraction are exclusively due to the Fe doping of the size-dependent crystallite of the Cr_2_O_3_ structure associated with movements of Fe/Cr ion along the *z*-axis with the Wyckoff position (0,0,z), showing a transition moment parallel to the hexagonal *c*-axis and affecting the long Fe–O bonds [82] directly.

This observed Fe-doping-dependent size increase in the structural properties of Cr_2_O_3_ discussed above is reflected in the electronic properties. In this context, the nominal valences of the cations in the Cr_2−*x*_Fe*_x_*O_3_ (0.0 ≤ *x* ≤ 0.18) nanoparticles were calculated considering the valence and fraction of each ion for each cation site. We employed the bond valence sum (BVS) analysis method [83,84]. This model, which is based on Pauling’s concept of electrostatic valence, describes that the sum of bond valence *S_ij_* around any ion, *i*, which is equivalent to the valence, *V_i_*, of this ion, and can be computed by the following equation:(4)$$V_{i}=\underset{j}{\sum }S_{ij}=\underset{j}{\sum }exp\left[\frac{\left(R_{0}-R_{ij}\right)}{B}\right]$$
where the sum is over all neighboring atoms *j* of the atom *i*. Here, *B* is an empirically determined “universal” constant equal to 0.37 Å [83]. *R_0_* represents the tabulated length of a bond of unit valence, taken to be 1.724 and 1.734 Å for Cr^3+^–O^2−^ and Fe^3+^–O^2−^ ion pairs, respectively [83], and *R_ij_* is the experimentally determined distance between atoms *i* and *j* in the coordination environment around the metal. The BVS analysis of the structure of the Cr_2−*x*_Fe*_x_*O_3_ (0.0 ≤ *x* ≤ 0.18) nanoparticles was performed using the Bondstr package incorporated in FullProf [85], and the results are given in Appendix A. As can be seen, the calculated BVS values for O ions are slightly closer to the formal valence of 2−, being between 1.935 < O^2−^ < 2.084. Moreover, the BVS values for Fe and Cr are reasonably consistent with the formal valence of 3+. However, increasing the Fe doping from 0 to 0.18 results in the Cr BVS value increment from 2.903 to 3.345, while the BVS value for Fe decrease from 3.609 to 3.126 of *x* = 0.06 to *x* = 0.18. As can be seen in Figure 2a, Fe^3+^ replaced Cr^3+^ sites in the same 6-fold coordination of the Cr_2_O_3_ host lattice, resulting in a <Cr–O> bond longer than the <Fe–O> bond. On this basis, we considered the anisotropic contractions of the unit cell, as followed by a significant decrease in the *c*-axis length relative to the a-axis, which results in movements of the Fe/Cr ion along the *z*-axis with the Wyckoff position (0,0,z), likely inducing a transition moment parallel to the hexagonal *c*-axis [82]. Accordingly, the Fe–O bond is strengthened, whereas the higher electron density of Fe and O results in enhanced covalency in the Fe–O bond. Thus, the valence state of the Fe is correlated with a lesser degree of hybridization between the Cr 3*d* and O 2*p* orbitals because of the longer Cr–O bond length.

We also used MEM and DFT to further characterize the effect of Fe doping on the modification of the Cr–Cr electrostatic repulsions and Cr–O bonding covalency character in Cr_2_O_3_. Figure 4a shows the two-dimensional (2D) electron density distribution on the (110) and (001) planes for the Cr_2−*x*_Fe*_x_*O_3_ (0.0 ≤ *x* ≤ 0.18) nanoparticles. The (110) and (001) planes are situated at 4.30 and 3.35 Å from the origin of the primitive cell, respectively. The 2D maps reveal that although the cation coordination polyhedral contains nearly regular octahedrons, the effect of Fe doping on the electron distribution is anisotropic. For Cr_2_O_3_, the characteristic Cr–O bonds in the (110) and (001) planes are both positive and of comparable magnitudes (0.01 and 29.0 *e*/Å^3^), indicating a significant σ-type covalent character for this bond due to both spin-up and spin-down electrons. However, the perfectly polar or ionic nature of the Fe/Cr–O bond is observed for the Fe-doped Cr_2_O_3_. The Cr–O bonds get more ionic in the (001) planes (‖c) while progressively developing a polar-covalent character in the (001) planes (⟂c) as the Fe doping content increases. This behavior can be explained by considering the non-centrosymmetric configuration with the C3v point symmetry of the metal octahedral environment exhibited in Cr_2_O_3_.

Indeed, the Cr_2_O_3_ crystal lattice is described as chains of octahedral sites constituted by [Fe_2_O_9_] dimers directed along the *c*-axis and separated from each other by an empty octahedral site (Figure 4b), whereas two face-sharing Cr^3+^ octahedral sites are present. Then, the common face area of two octahedra forming a dimer is smaller than that of the faces sharing an empty cationic site because only the two octahedral triangular faces perpendicular to the c-axis remain equilateral. There is a natural C3v-type distortion of the [CrO_6_] cages, resulting in the rhombohedral unit cell flattened along the *c*-axis and the face-sharing octahedral site. Additionally, the Cr is displaced from the geometric octahedral site center along the *c*-axis towards the large equilateral face of the [CrO_6_] octahedra, forming two sets of Cr–O bond distances. Thus, the study of the formation of three short bonds between the ligands of the sizeable equilateral face and the metallic center and three long bonds between the small equilateral face and the metallic center (Figure 4b) is valuable when inspecting the Cr–Cr repulsion across the shared-face octahedra. The electron density values at critical bond points for all samples are presented in Table 1, which confirms and quantifies the decrease in the bonds’ covalent strength with the Fe doping. Although the mid-bond electron density values for the short (Cr,Fe)–O bond did not fall in a consistent trend, the long (Cr,Fe)–O bond strengthened with the Fe doping, as evident from the increase in the mid-bond electron density from 0.2273 *e*/Å^3^ to 0.7469 *e*/Å^3^ as *x* = 0.12. Nevertheless, this value was decreased to 0.3431 *e*/Å^3^ for x = 0.18, further indicating the dynamic equilibrium state of the Cr^3+^ electronic structure in the axially compressed CrO_6_ octahedra. These results suggest that the electron density enhancement for the samples with *x* = 0.06 and 0.12 may come from an existing electron transfer among the Cr sites in the Cr_2_O_3_ lattice, as induced by the higher reducing ability and electronegativity of Fe (Fe = 1.83 > Cr = 1.66) [86]. Then, by substituting Cr^3+^ with Fe^3+^, the higher reducing ability and electronegativity favor Fe-containing sites that attract O atoms more strongly, shortening the Fe–O distances and affecting that equilibrium state with the increase in Fe content. In contrast, a lower electron density in the O-planes is expected to increase the polar strength along the *c*-axis, resulting in the orbital polarization of *d^3^* electrons on Cr by the Fe effect in the crystalline environment, as indicated by the strongly directional lobes consistent with the spatial orientation of *t_2s_* orbitals present in Figure 4a. This suggests that majority-spin electrons are expected to show antibonding to balance the bonding effect of those with a minority-spin nature, decreasing the covalent contribution and introducing Cr^3+^ sites surrounded by electron-rich structures. Furthermore, the higher usual effects of shrinking the electron density on the Fe^3+^/Cr^3+^ and O^2−^ sites (see Figure 2a) was evident for *x* = 0.18, suggesting that the lattice constraints were relaxed, allowing the Madelung energy to be optimized by increasing the strongest Cr–Cr interaction. This indicates that the critical distance of the Cr–Cr interaction is achieved for doping contents around the Fe doping mid-point (herein, *x* = 0.12). In this place, it is also worth mentioning that although negative characters corresponding to exchange repulsion are observed for O–O pairs due to longer Fe–O bonds, the associated Cr–Cr interaction because of the Madelung electrostatic field, and exchange repulsion, the possible bond overlap populations for Cr–Cr pairs are negligible up to *x* = 0.18, indicating that no metal bonding is present, which is fully consistent with the DOS of Cr_2_O_3_ showed in (Figure 6).

We further studied the nature of the Fe/Cr–O bonding covalency in Cr_2_O_3_ via DFT. The optimized lattice parameters *a* = *b* and *c*, the critical bond points, the Bader charges q(Ω), and the results of the BVS simulated data are summarized in Table 2. As can be observed, the computationally optimized *a* = *b* = 5.005 Å and *c* = 13.626 Å parameters describe a Cr_2_O_3_ system with a unit cell volume that is 2.2% greater than the experimental result shown in Appendix A. For Fe-doped cells, the optimized volume also changes, but the result corresponds to a unit cell with a 1.9% smaller volume. This behavior is interesting because the relaxed cell shows an increase along the *c*-axis with a decrease along the *a*- and *b*-axes, which leads to an axial ratio of *c*/*a* = 2.808 that is greater than in the pure case (*c*/*a* = 2.722). When compared to the experimental systems, Appendix A shows that the unit cell volume practically does not change as the percentage of Fe dopants within the Cr_2_O_3_ lattice increases. Despite this, it is possible to note a slight tendency to increase the cell along the *c*-axis, which corroborates the theoretical findings. Further, our simulated cell parameters are consistent with other studies carried out with B3LYP [70] and GGA+U [51] exchange–correlation functionals. Changes and interpretations of chemical bonds as well as the charge state estimations of the removed (Cr) and inserted (Fe) ions were calculated in connection with Bader’s topological procedure implemented in the Critic2 computer program [87,88]. For clarity’s sake, the results displayed in Table 2 for Cr–O and Fe–O refer to bonds with non-equivalent oxygen atoms that form their respective local structures (Figure 5). Each of these oxygen atoms, which are named O1 and O2, has a cell multiplicity equal to 3, leading to the conclusion that the octahedral environment around the Cr sites in the Cr_2_O_3_ crystalline matrix is preserved after the insertion of the Fe impurity. This is coherent with the experimental discussion presented before. The electronic density values at the bcps (ρb) demonstrate an interesting phenomenon; while the presence of the Fe defect causes a slight increase in the bond length going from Cr–O1 to Fe–O1, an opposite effect can be perceived going from Cr–O2 to Fe–O2. This reveals that Fe ions induce a decrease in the degree of the covalent nature along with the Fe–O1 bonds and an increase along with Fe–O2 bonds.

The electron density map illustrated in Figure 5 helps us to visualize these changes in the chemical behavior. We consider the density in the crystal plane due to the replaced chromium (Cr_2_O_3_) and inserted iron (Cr_2_O_3_: Fe) and its O1 and O2 ligands, which represent their 6-fold coordination. Evidently, the charge density around the Fe site is more symmetrical than around chromium. The reason for this comes from the fact that the Fe’s *d*-orbitals are occupied with a greater number of electrons than in the Cr ions’ case. Moreover, we note that the charge density overlaps more along with Fe–O2 than with Cr–O2 bonds. From the magnetic perspective, this means that the exchange interaction acts in such a way that it favors the antiferromagnetic order along Fe–O2 at the same time, tending to disfavor it along with the Fe–O1 bonds. The calculated atomic Bader’s charges *q*(Ω) for Cr, Fe, and their O1 and O2 ligands are shown in Table 2. According to the results, when the Fe defect is incorporated into the compound, an equal oxidation state to the substituted Cr ion is assumed, while the charge states of the O1 and O2 oxygen atoms practically do not change. These conclusions agree with the fact that the Fe dopant has a local geometry like that of the Cr atom in a perfect crystal. The difference between the calculated and formal oxidation states corresponding to Cr^3+^, Fe^3+^, and O^2−^ is not surprising because the atomic charges obtained from the Bader procedure are normally smaller than those determined experimentally.

The BVS depicted at the end of Table 2 was obtained based on fully PBEsol-relaxed parameters of the pure and Fe-doped structures. The BVS model is often used to assess information about the plausibility of the crystal structure and to understand the chemical bonding by comparing its values with the formal valence state. As can be seen, our simulated BVS is close to the expected formal valences for Cr, Fe, and O ions. The highest BVS values for Fe and its O1 and O2 ligands are consistent with their shortest bond lengths. Comparing these with the experimental BVS results reported in Appendix A, we can see that they are in good agreement; the values for the perfect Cr_2_O_3_ crystal are almost the same, and the Fe-containing structure shows the same upward trend. These results reinforce our conclusions about the chemical and structural modifications caused by the insertion of the Fe dopant in the host matrix.

To understand the electronic properties of the Cr_2_O_3_ and its Fe-doped counterparts, their total and partial spin-polarized density of electronic states (TDOS and PDOS, respectively) were calculated as illustrated in Figure 6. The valence band (VB) maximum and conduction band (CB) minimum are composed essentially by the Cr *d*-orbitals (Figure 6a,b). Notably, the O *p*-states occupy mostly lower energy regions, but a small hybridization area with Cr *d*-orbitals at the top of the VB and the bottom of CB can be noticed. Our calculated electronic band gap of 3.38 eV is very close to the experimental value (~3.4 eV) [90,91]. This is a notable improvement of the bandgap description compared to the other published studies with hybrid (B3LYP) [70] or combined GGA+U functional schemes [51]. Firstly, because our outcome describes electronic bands that correctly corroborate with the Mott–Hubbard insulator results and charge transfer behavior founding for this oxide material, and secondly because the TB-mBJ XC semi-local potential, which we use efficiently, describes the electronic properties of the compound with a lower computational cost, since B3LYP makes use of calculations involving dual-center integrals and GGA+U, the on-site Hubbard parameter. Additionally, the TDOS and PDOS plots (Figure 6c,d) for the Fe-doped Cr_2_O_3_ system show that the Fe ion introduces its d-states at well-localized regions into the bandgap. The basic shape and orbital character of VB and CB do not change significantly. The most apparent difference is the highlighted band positioned approximately between −2 and −1 eV, arising from the superposition of the Fe’s *d*-orbitals with the *p*-states of its nearest-bound oxygen atoms. This indicates that for low Fe doping concentrations, Fe atoms are expected to become isolated, and the Fe 3*d* bands most likely transform into a set of isolated states. Furthermore, the unoccupied Cr 3*d* band tends to move towards higher energy levels as the Fe doping increases. Accordingly, the oxygen-intermediated Fe-Cr interactions become larger because the spin polarization on the Cr^3+^ ions will adopt a configuration like that of α-Fe_2_O_3_.

The MEM and DFT results showed some similarities between them, such as the fact that the Cr–Cr distance might play a role in the covalence of the Cr–O bond by broadening the 3*d* band. The Cr–O distances in Cr_2_O_3_ and Fe-doped Cr_2_O_3_ nanoparticles are likely to close, since the studied concentration of iron was very low. However, a greater Fe–O bonding covalency in Fe-doped Cr_2_O_3_ than for Cr–O in Cr_2_O_3_ was observed. Although the electron transfer concerns the *e_g_* states and is negligible for the *t_2g_* ones, the higher electron density connecting the Fe–O bonding signals that Fe-incorporation-dependent variations on the Cr–Cr distance induce a modification of the electrostatic repulsions, favoring changes in the bond mechanism of the three outermost electrons of the *t_2g_* symmetry, which are responsible for metal–metal covalent (1e^−^) and Cr–O (2e^−^) bonds. Thus, higher Fe doping results in negative increases in the potential energy level toward the band-edge conduction due to the Fe affinity for oxygen. Furthermore, it is expected that by increasing the Fe doping content in Cr_2_O_3_, an asymmetric PDOS distribution of the electronic spin polarization will reveal split energy bands, and the atomic planes of the mixed Cr–Fe interface will adopt the ferromagnetic coupling structure in Fe systems doped with Cr_2_O_3_. This is expected due to the strong antiferromagnetic coupling between Cr^3+^ and Fe^3+^ in corner-sharing octahedra from neighboring layers [51]. Thus, the three unpaired electrons in Cr fully filled *t_2g_* orbitals result in a near cubic shape of the spin density distribution. In contrast, a near-spherical spin density distribution is expected because the five unpaired electrons in Fe that are fully filled are both *t_2g_* and *e_g_* orbitals. This supports the assumption that a strong electron correlation (the Mott–Hubbard correlation) effect mediated by a weak cation–cation interaction is responsible for the Fe-doping-induced changes in magnetic properties of the Cr_2_O_3_ nanoparticles, as is well-known for corundum-like structure metal oxides [92,93]. Moreover, it is worth highlighting that the spin density distributions at the Cr/Fe sites induce a complex character for the spin polarization at the oxygen sites, contributing to O 2*p* band broadening and the formation of a complex pattern of Cr/Fe−Cr/Fe spin interactions for both MEM and DFT results. Therefore, the Cr_2_O_3_ and Fe^3+^-doped Cr_2_O_3_ nanoparticles can be thought of as Mott–Hubbard insulators, whereas the band gap is in the *d*−*d* gap, and the conduction band is dominated by Cr 3*d* orbitals. Finally, our results also point to the fact that the magnitude and the character of the electronic density near the O atom bonds in Cr_2_O_3_ nanoparticles can vary throughout the Fe^3+^ doping, being modulated by the value of the Cr–Cr distances until reaching stabilization in the quasi-equilibrium structure for Fe doping values higher than *x* = 0.12. This might have consequences for the use of Fe-doped Cr_2_O_3_ as a possible candidate in the photochemistry industry, and reaction catalysis is an excellent option for magneto-optoelectronic device applications.

## 4. Conclusions

In summary, using the maximum entropy method (MEM) applied to X-ray diffraction data and density functional theory (DFT) calculations, we carefully investigated the Fe^3+^ doping effects on the structure and electron distribution of Cr_2_O_3_ nanoparticles. We showed that increasing the Fe^3+^ doping content induces an enlargement in the axial ratio of *c*/*a*, which is associated with an anisotropic expansion of the unit cell. Moreover, we demonstrated that as Fe^3+^ replaced Cr^3+^ in the Cr_2_O_3_ lattice, this caused a greater interaction between the metal 3*d* states and the oxygen 2*p* states, leading to a slight increase in the Cr/Fe–O1 bond length followed by an opposite effect for the Cr/Fe–O2 bonds. Our results also suggested that the Fe dopant introduces *d*-states at a well-localized bandgap region, resulting in the strong covalent nature of the Fe–O2 bonds compared to Fe–O1 bonds. Moreover, we found that the band gaps in the mixed systems are characterized by excitations from occupied Cr *d* to unoccupied Fe *d* states. The Cr_2_O_3_ and Fe-doped Cr_2_O_3_ nanoparticles behave as Mott–Hubbard insulators, whereas their band gap is in the *d*−*d* gap and the conduction band are dominated by Cr 3*d* orbitals. Furthermore, the magnitude and the character of the electronic density near the O atom bonds in Cr_2_O_3_ nanoparticles are modulated by the value of the Cr–Cr distances until reaching stabilization in the quasi-equilibrium of the Cr_2_O_3_ lattice for Fe doping values in the saturation level range. These findings suggested that Fe-doped Cr_2_O_3_ can be considered a promising candidate for spintronics and magneto-optical devices.

## Figures and Tables

**Figure 1 nanomaterials-13-00980-f001:**
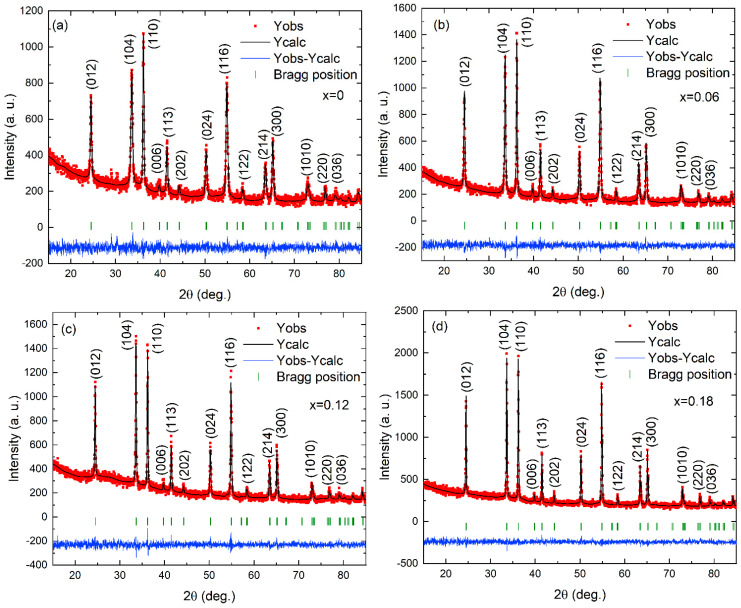
Observed (red), calculated (black), and differential patterns of Rietveld refinements of X-ray diffraction for Cr_2−*x*_Fe*_x_*O_3_ nanoparticles: (**a**) *x* = 0, (**b**) *x* = 0.06, (**c**) *x* = 0.12, and (**d**) *x* = 0.18. The red square symbols and the black line denote the observed and calculated intensities, respectively. Short vertical lines (green) indicate the positions of the possible Bragg reflections of the rhombohedral structure. The difference between the observed and calculated profiles (blue) is plotted at the bottom.

**Figure 2 nanomaterials-13-00980-f002:**
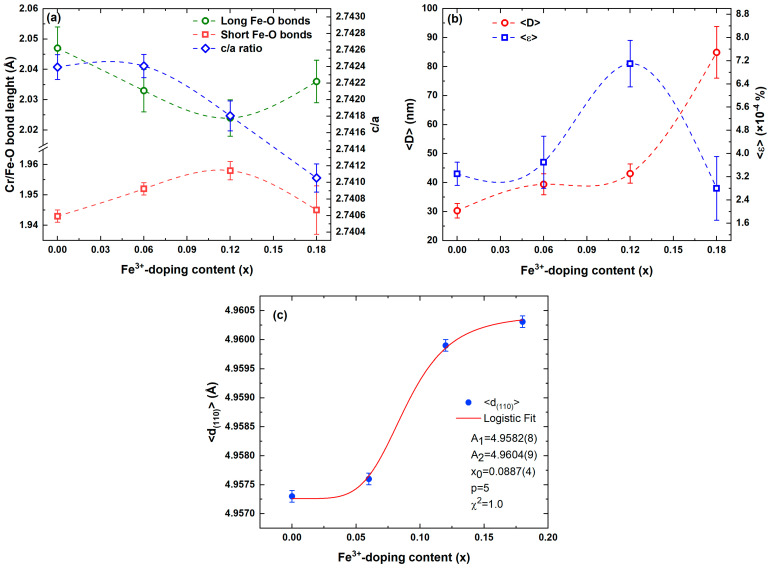
(**a**) Fe–O bond length and *c*/*a* ratio, (**b**) average crystallite size (<*D*>) and microstrain (<ε>), and (**c**) Cr–Cr distance in the (110) plane for the Cr_2−*x*_Fe*_x_*O_3_ (0.0 ≤ *x* ≤ 0.18) nanoparticles.

**Figure 3 nanomaterials-13-00980-f003:**
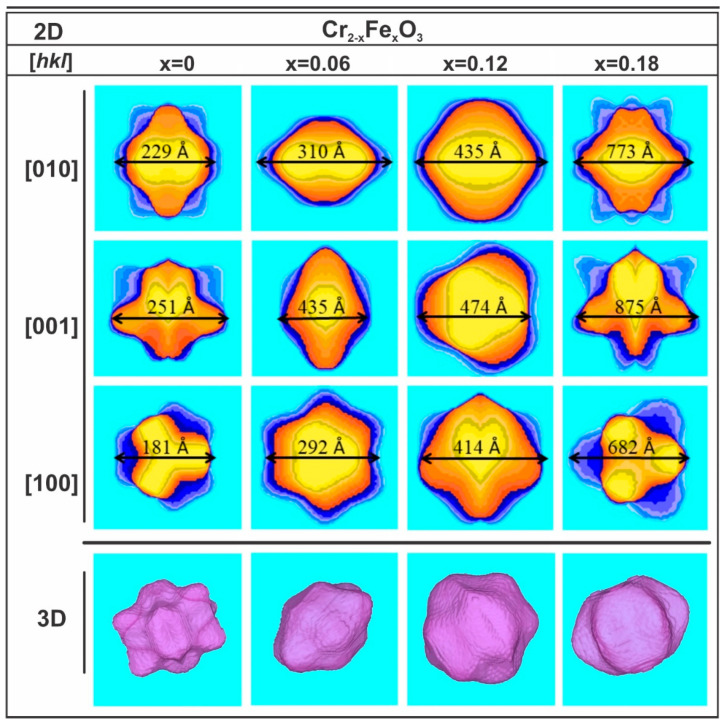
The 2D and 3D apparent crystallite size projections for Cr_2−*x*_Fe*_x_*O_3_ (0.0 ≤ *x* ≤ 0.18) nanoparticles along the [010], [001], and [100] crystallographic directions.

**Figure 4 nanomaterials-13-00980-f004:**
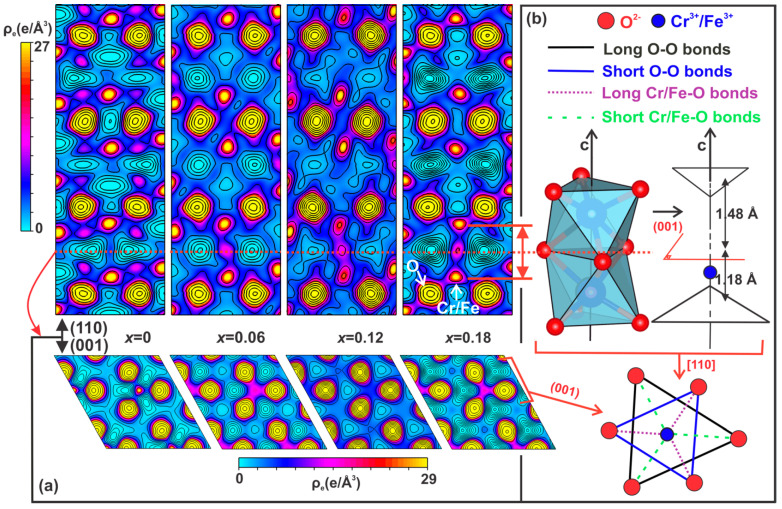
(**a**) Two-dimensional (2D) electron density maps drawn parallel to he (001) and (110) crystallographic planes obtained from MEM analyses for the Cr_2−*x*_Fe*_x_*O_3_ (0.0 ≤ *x* ≤ 0.18) nanoparticles. (**b**) Structural representations of chains of [CrO_6_] dimers along the *c*-axis in the Cr_2_O_3_ structure, highlighting the small and large equilateral triangles and Cr^3+^ displacement from the octahedral site center (upper right-side sketch). The sketch below represents the two face-sharing FeO_6_ octahedra, as shown in the upper right-side sketch, whereas there are three short and three long nearest-neighbor Fe/Cr–O distances inside each [Fe/Cr]O_6_ octahedron.

**Figure 5 nanomaterials-13-00980-f005:**
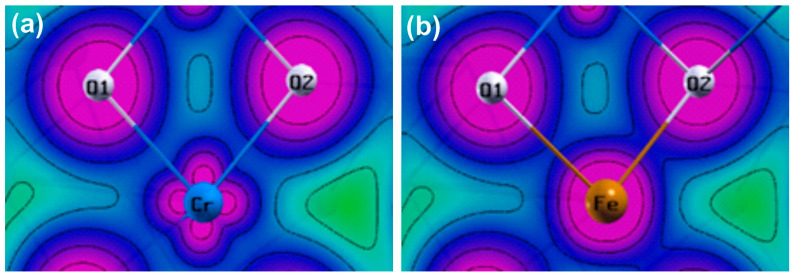
(**a**) Total valence electron density of Cr_2_O_3_ and (**b**) Fe-doped Cr_2_O_3_ crystals generated in the crystal plane through Cr and Fe ions to evidence the chemical bonding changes of the corresponding local structures. Produced using Xcrysden software [89].

**Figure 6 nanomaterials-13-00980-f006:**
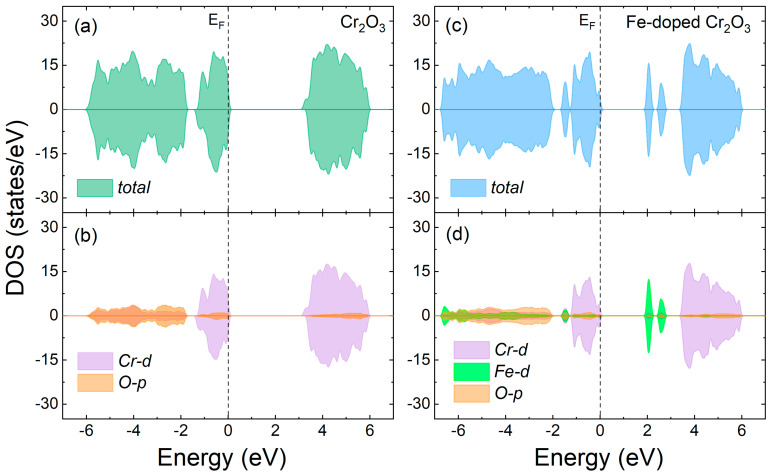
Total (TDOS) and projected spin-polarized density of electronic states (PDOS), as calculated using TB-mBJ exchange–correlation potential: (**a**,**b**) the TDOS and PDOS of Cr_2_O_3_, respectively; (**c**,**d**), the TDOS and PDOS for Fe-doped Cr_2_O_3_, respectively. The dashed black line is the Fermi energy level, and the positive and negative values represent the states with the spin polarization moving up and down, respectively.

**Table 1 nanomaterials-13-00980-t001:** The calculated bond lengths and mid-bond electron density for short and long Cr/Fe–O bonds along the (001) and (110) planes and the Cr–Cr distance in the (110) plane for the Cr_2−*x*_Fe*_x_*O_3_ (0.0 ≤ *x* ≤ 0.18) nanoparticles.

Fe Doping (x)	(Cr,Fe)–O Short		(Cr,Fe)–O Long		Angle (Deg.)	Cr–Cr Distance (Å)
	Bond Length (Å)	Mid-Bond Electron Density (e/Å3)	Bond Length (Å)	Mid-Bond Electron Density (e/Å3)		
0	1.943(2)	0.43445	2.047(9)	0.22747	80.3(4)	4.95730(1)
0.06	1.952(2)	0.36455	2.033(7)	0.54572	80.8(3)	4.95760(1)
0.12	1.958(3)	0.52716	2.024(6)	0.74691	82.5(4)	4.95990(1)
0.18	1.945(8)	0.51283	2.036(7)	0.34307	82.9(1)	4.96031(1)

**Table 2 nanomaterials-13-00980-t002:** Calculated properties based on structural (lattice constants and BVS data) and electronic (Bader’s charges and critical bond points) analyses for the Cr_2_O_3_ and Fe-doped Cr_2_O_3_ compounds. The O1 and O2 oxygens indicate the non-equivalent ligands, which compose the local structure around the Cr site and of the Fe-defect.

Cr_2_O_3_			Fe-doped Cr_2_O_3_		
Optimized Lattice Parameters					
*a* = *b* (Å)	5.005		4.885		
*c* (Å)	13.626		13.721		
*c*/*a*	2.722		2.808		
Volume (Å^3^)	295.682		283.591		
Bond Critical Points					
Bond length (Å)		ρ*_b_*(*e*/Å^3^)	Bond length (Å)		ρ*_b_*(*e*/Å^3^)
Cr–O1	2.038	0.072	Fe–O1	2.098	0.070
Cr–O2	1.980	0.082	Fe–O2	1.958	0.092
Bader charges *q*(Ω)					
Cr	+2.0		+2.0		
Fe	-		+2.0		
O1	−1.35		−1.35		
O2	−1.38		−1.37		
Bond Valence Sum (BVS)					
Cr	2.86		2.86		
Fe	-		2.94		
O1	1.89		2.06		
O2	1.94		2.14		

## Data Availability

The data that support the findings of this study are available from the corresponding author upon reasonable request.

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
