# Peer review of "Insights into the Fe3+ Doping Effects on the Structure and Electron Distribution of Cr2O3 Nanoparticles"

_nanomaterials, 2023, doi:10.3390/nano13060980_

Round 1

Reviewer 1 Report

The authors provide a very rigurous and complete study of Cr2O3 nanoparticles combining experimental XRD measurements with DFT and MEM calculations to assess the effect of Fe-doping on the structural and electronic properties of the particles.

The methodology is appropriate, the results clearly explained and the findings very relevant, namely they find a direct link between Fe-doping and unit cell anisotropic expansion, as well as the strength of the bonds by modulating electronic states. 

The authors could maybe add some perspectives and mention how this study can serve as reference on other specific applications

They should also replace Paulin's by Pauling's

Reviewer 2 Report

In this article, the authors investigate theoretically and experimentally the incorporation of Fe3+ in Cr2O3 nanoparticles. From the existing literature it can be concluded that this work is original and deserves publication  The article is well-written and contains important findings for this kind of particles regarding structure and conductivity modification after the introduction of Fe3+ in the lattice. The conclusions are supported by the results and the initial goal regarding the effects of introducing iron in the nanoparticles is successfully achieved. However, I would like the authors explain what is the  evidence of the valence of Fe within the nanoparticles, and finally of the composition of the nanoparticles, beyond the nominal expected values. I can understand that Rietveld refinement of the XRD spectra can give information about the electon density in the cell but experimantally, XPS would give some supporting information regarding bonding and valence of the ions involved. It would be useful such an indormation if possible.

I suggest publication and perhaps if the authors can do such an experiment XPS  would be useful for the copmletness of this iteresting subject.

Reviewer 3 Report

Doping effects on Cr2O3 nanoparticles are reported with the aid of DFT solid

state computations, decorated by some experimental studies.

Several properties are studied and the authors reach to some conclusion as far as concerns the alteration of the physical/electronic characteristics upon doping.

Two issues are missed or nor properly highlighted,

A. The novelty of the reported results and how these could contribute to the understanding of the electronic structure of Cr2O3 with different 229
Fe-doping content.

B. Any connection with the experimetal data is missed or partially provided. How the computational data could assist the evaluation of reported crystal data ? How the alteration of the electronic structure character, upon doping, could affect the optical properties of the crystal? Why the authors select to study the charge density changes and not other related properties ?

C. Have the authors employed spin polarized DFT method or not ?

Round 2

Reviewer 3 Report

The authors properly address the issues raised by the reviewers.

A Minor issue to enrich the conclusion section with some potential application(s).

Author Response

INCORPORATE--become an actual part of the text:

These findings suggested that Fe-doped Cr2O3 as can be considered a promising candidate for spintronics and magneto-optical devices.